# Label-Free Optical Transmission Tomography for Direct Mycological Examination and Monitoring of Intracellular Dynamics

**DOI:** 10.3390/jof10110741

**Published:** 2024-10-26

**Authors:** Eliott Teston, Marc Sautour, Léa Boulnois, Nicolas Augey, Abdellah Dighab, Christophe Guillet, Dea Garcia-Hermoso, Fanny Lanternier, Marie-Elisabeth Bougnoux, Frédéric Dalle, Louise Basmaciyan, Mathieu Blot, Pierre-Emmanuel Charles, Jean-Pierre Quenot, Bianca Podac, Catherine Neuwirth, Claude Boccara, Martine Boccara, Olivier Thouvenin, Thomas Maldiney

**Affiliations:** 1Lipness Team, Translational Research Center in Molecular Medicine– INSERM Joint Research Unit (CTM-UMR1231), University of Burgundy, 21000 Dijon, France; 2Department of Parasitology/Mycology, Dijon Bourgogne University Hospital, 21000 Dijon, France; 3Unité mixte de recherche Procédés Alimentaires et Microbiologiques (UMR PAM) A 02.102, Bourgogne Franche-Comté University, AgroSup Dijon, 21079 Dijon, France; 4Medical Biology Laboratory, William Morey General Hospital, 71100 Chalon-sur-Saône, France; 5LISPEN, Arts et Metiers Institute of Technology, 71100 Chalon-sur-Saône, France; 6Translational Mycology Research Group, Mycology Department, National Reference Center for Invasive Mycoses and Antifungals, Institut Pasteur, Paris Cité University, 75015 Paris, France; 7Necker-Enfants Malades Hospital, Assistance Publique-Hôpitaux de Paris, 75015 Paris, France; 8Infectious Diseases Department, Dijon Bourgogne University Hospital, 21000 Dijon, France; 9Department of Intensive Care Medicine, Dijon Bourgogne University Hospital, 21000 Dijon, France; 10Department of Bacteriology, University Hospital of Dijon, 21000 Dijon, France; 11UMR/CNRS 6248 Chrono-Environnement, Bougogne Franche-Comté University, 25000 Besançon, France; 12Institut Langevin, Ecole Supérieure de Physique et Chimie Industrielle de la ville de Paris, Université PSL, CNRS, 75005 Paris, France; 13Institut de Systématique, Evolution, Biodiversité– (ISYEB-UMR7205), Ecole Normale Supérieure, PSL Research University, 75005 Paris, France; 14Department of Intensive Care Medicine, William Morey General Hospital, 71100 Chalon-sur-Saône, France

**Keywords:** optical tomography, medical mycology, invasive fungal infections, live-cell imaging, fungal metabolism

## Abstract

Live-cell imaging generally requires pretreatment with fluorophores to either monitor cellular functions or the dynamics of intracellular processes and structures. We have recently introduced full-field optical coherence tomography for the label-free live-cell imaging of fungi with potential clinical applications for the diagnosis of invasive fungal mold infections. While both the spatial resolution and technical set up of this technology are more likely designed for the histopathological analysis of tissue biopsies, there is to our knowledge no previous work reporting the use of a light interference-based optical technique for direct mycological examination and monitoring of intracellular processes. We describe the first application of dynamic full-field optical transmission tomography (D-FF-OTT) to achieve both high-resolution and live-cell imaging of fungi. First, D-FF-OTT allowed for the precise examination and identification of several elementary structures within a selection of fungal species commonly known to be responsible for invasive fungal infections such as *Candida albicans*, *Aspergillus fumigatus*, or *Rhizopus arrhizus*. Furthermore, D-FF-OTT revealed the intracellular trafficking of organelles and vesicles related to metabolic processes of living fungi, thus opening new perspectives in fast fungal infection diagnostics.

## 1. Introduction

Fungal infections represent an important class of infectious diseases, ranging from cutaneous dermatophyte infections to more invasive fungal infections (IFIs), responsible for significant morbidity and mortality worldwide [1,2]. Due to either a lack of an appropriate diagnostic tool, the slow and fastidious growth of certain fungi, or sometimes a lack of trained biologists with sufficient experience in medical mycology, some IFIs still remain underdiagnosed [3,4]. Yet, IFI incidence is increasing in several Western countries and the potential role of novel convenient and affordable routine technologies may help in optimizing fungal infection diagnosis based on additional structural or metabolic information regarding specific fungal elementary structures, dynamic processes, and the development of automated methods [5].

So far, the visualization of fungal developmental processes relies on standard scientific methods that allow for the precise monitoring of intracellular structures and dynamics [6,7], with potential applications in the development and testing of antifungal molecules. Among them, fluorescence or confocal laser scanning microscopy (CLSM) associated with specific fluorescent moieties targeting different cellular compartments appears as the most versatile and convenient technique to observe fungi [6]. Many developed fluorescent dyes are commonly used as vital organelle markers for filamentous fungi and present the advantage of allowing imaging at relevant concentrations lower than their toxic threshold [7]. However, the transposition of such techniques to daily practice in clinical microbiology is hampered by staining and time constraints. Recent mass spectrometry or molecular-based diagnosis technologies offer faster alternatives, but these techniques still require costly pre-assay processing, which may lead to false negatives [8]. In addition, as the time to diagnosis is a major prognosis factor of mortality for many systemic mycoses [9,10,11], the development of alternative automated diagnosis techniques allowing for the subsequent use of machine learning and artificial neural networks would be of precious help to envision on-site IFI diagnosis [12].

To this end, our group recently introduced the use of dynamic full-field optical coherence tomography (D-FF-OCT) as a valuable tool for the visualization of microscopic filamentous fungi from solid cultures [13]. While this technique allowed for the observation of different molds’ structures and metabolic activity (hyphae, spores, conidia), in view of the potential histopathological identification of fungi within tissue biopsies, its spatial resolution and optical set up were not suitable for direct cytological examination. We therefore developed a complementary interferometric approach intended for label-free live-cell imaging, called dynamic full-field optical transmission tomography (D-FF-OTT), based on a transmission and self-referenced configuration, allowing for a higher spatial resolution than D-FF-OCT (Aquyre Biosciences, Inc, États-Unis) and the subsequent dynamic imaging of biosystems [14,15]. Despite previous applications for the visualization of diatoms, ex vivo biological samples (stromal and endothelial cells), and a successful attempt to monitor HeLa cell metabolism, there is to our knowledge no report of this technology in medical mycology. The present contribution describes the first use of D-FF-OTT for the direct examination of important fungal strains, among which are pathological yeasts, filamentous, and Mucorales strains [16].

## 2. Materials and Methods

### 2.1. Fungal Strain Sample and Culture

The following fungal strains were used for all experiments: *Candida albicans* ATCC 90028, *Candida parapsilosis* ATCC 22019, *Aspergillus fumigatus* ATCC 204305, and wild-type clinical strains of *Aspergillus niger* and *Rhizopus arrhizus*. Depending on the fungal strain, solid cultures were grown following incubation times varying between 1 and 7 days at 30 °C in a humidified chamber on Sabouraud dextrose agar (BioMérieux SA, Marcy-l’Etoile, France) before subsequent microscopic observation.

### 2.2. Microscopy

All microscopy experiments were conducted following the same procedure. Fungi were harvested realizing an adhesive tape preparation of samples, where a droplet of physiological serum or lactophenol blue was dropped on the glass slide and a piece of adhesive tape was used to harvest fungal cells and was then put in contact with the droplet. A coverslip (22 × 22 mm) was then added on top of the droplet, and fluorescence microscopy immersion oil was also added for high-magnification bright-field microscopy and FF-OTT.

#### 2.2.1. Bright-Field Microscopy

All strains were observed using a BX46 microscope (Olympus, Nagano, Japan) with or without prior lactophenol blue (RAL diagnostics, Martillac, France) staining.

#### 2.2.2. D-FF-OTT

Lactophenol blue-stained and unstained fungal cultures were imaged using a dedicated prototype of a D-FF-OTT-based microscope. As described in Figure 1, this D-FF-OTT prototype system uses a focused incoherent λ = 455 nm light-emitting diode (LED) (M455L4, Thorlabs, Newton, NJ, USA) with a 100× achromatic microscope objective (A100X-V300, Amscope, United Kingdom), an achromatic tube lens (AC254-300-A-ML, Thorlabs, Newton, NJ, USA), and a complementary metal oxide semiconductor (CMOS) MV1-D1024E camera (Photonfocus, Lachen, Switzerland) to magnify and record the sample plane on the camera.

Figure 1 shows the OTT prototype that is used in this study. Figure 1a shows the global set up, while Figure 1b shows a detailed schematic of this prototype while observing *Candida albicans* (images on schematic computer screen). Figure 1c shows a zoom-in of the objective area of the OTT system.

As the physical phenomenon underlying this technique is light interferences, the adjustment of the position of the objective leads to the observation of a given slice of the sample that is being looked at. Indeed, the particularity of optical transmission tomography techniques is that the thickness of the optical section is independent of the depth of field, as it is determined by the spectral bandwidth of the light source. In our situation, the axial resolution is around 1 to 2 μm [14].

### 2.3. D-FF-OTT Acquisition

As represented in Figure 1d, during D-FF-OTT microscopy, a series of 128 successive frames were acquired at a frame rate of 100 fps. After the acquisition, this temporal stack was processed to form the corresponding HSV colored D-FF-OTT image that recapitulates the local fluctuations [14,18]. This color image is based upon the 3 following independent channels:Hue, or the central color, measures dynamic fluctuations (how fast does the D-FF-OTT signal fluctuate). For each pixel, the absolute value of the Fourier transform of the normalized time series is calculated. The cumulative distribution of half the spectrum is calculated and the median value defines the hue of the corresponding pixel.Saturation, or the purity of each color, measures the frequency range. The 25th and 75th percentiles from the previous distribution are measured to extract the frequency bandwidth (where 50% of the energy of the fluctuation is).Value, or the brightness of each color, measures the amplitude of dynamic fluctuations. In practice, we subtract the mean value of the time series and divide it into buckets of 8 images. The absolute value of the maximum value of the cumulative sum in each bucket is calculated, and the sum of these values finally gives the value of the corresponding pixel. This cumulative sum strategy was shown to be more efficient for detecting small biased random displacements versus a random noise as standard deviation [19].

Each independent channel is saved as a 16 bit image before normalization, then is normalized and combined to form an 8 bit HSV color image. The values of hue are reverted and normalized between 0 and 0.66 (with 1% saturation) so that high frequencies (fast fluctuations due to moving organelles) appear red and low frequencies (slow fluctuations) appear blue. The values of saturation are reverted and normalized between 0 and 1 (with 1% saturation) so that large bandwidths (typical noise) appear with a low saturation and narrow frequency bandwidths appear more contrasted. The values of brightness are normalized between 0 and 1 (with 1% saturation) so that large fluctuations appear brighter than small fluctuations. All three images are combined and transformed to an RGB image using the hsv2rgb function in Matlab.

One color image is obtained from each sample, but the normalization factors depend on each sample. To allow comparisons between different samples, a common normalization factor of each channel type is calculated based on the 1st and 99th percentiles of all raw images.

### 2.4. Quantitative Analysis

The automatic segmentation of yeast cell structures (specifically the cellular membrane, nuclei membrane, and organelles) was executed using sophisticated machine learning models underpinned by YOLOv8 architecture (Appendix A), known for its effectiveness in detecting objects in real time and above all, for its simplicity to set up.

The process began by preprocessing the initial dataset, which consisted of high-resolution microscopic images of yeast cells. These images underwent several stages of preprocessing using a tailor-made Python script for our specific case. We applied transformations such as sharpening, grayscale conversion, adaptive histogram equalization (CLAHE), noise reduction, and mapping colors to improve the visibility of cell membranes. This process was performed in a few seconds using an i5 processor.

Two distinct convolutional neural network models were developed for targeted tasks. The first model was able to detect the outer cellular membrane and classify it as one of two yeast species; it utilized this capability to differentiate and categorize cellular membranes (*Candida albicans* or *Candida parapsilosis*). The second model was adapted to recognize and delineate internal structures of yeast cells (nuclear membranes and organelles). This model relied on a refined dataset generated by the first model, which isolated individual cells against a black background, thereby simplifying the identification and analysis of intracellular structures.

These two models were trained on images measuring 512 pixels by 512 pixels and the distribution of the dataset for training was, respectively, around 70% of the images for training, 20% for validation, and 10% for the test for both training sessions. The datasets consisted of 117 images for the first training set and 305 images for the second training set.

From this automatic segmentation, we recovered the raw data of detected pixels for each intracellular structure of the yeast to subsequently carry out quantitative analyses.

The training process was performed in approximately 25 min using an NDIVIA Tesla4 graphics card and carried out on Google Collab.

### 2.5. Statistical Analysis

For statistical analysis, we used the online GraphPad Outlier calculator to perform Grubbs’ test (https://www.graphpad.com/quickcalcs/grubbs1/, accessed on 24 September 2024) when necessary, and Prism 5 (GraphPad Software) to perform an unpaired *t*-test. We present the data as mean values (standard error of the mean, SEM). We considered a *p* value < 0.05 to be statistically significant.

## 3. Results

### 3.1. Comparison of D-FF-OTT to Conventional Optical Imaging

Five fungal strains were selected to assess D-FF-OTT capacity to image typical fungi responsible for IFIs, namely *Candida albicans*, *Candida parapsilosis*, *Aspergillus fumigatus*, *Aspergillus niger*, and *Rhizopus arrhizus*. All strains were imaged using conventional optical microscopy with or without lactophenol blue staining and D-FF-OTT.

Figure 2 shows 100× images of *Candida albicans* and *Candida parapsilosis* acquired with either optical microscopy or D-FF-OTT. Images from standard bright-field optical microscopy (Figure 2a) confirm the visualization of *Candida albicans*. Similar to conventional optical microscopy, FF-OTT allows a clear distinction of all structures inside *C. albicans* (Figure 2b); we distinguish nuclei and vacuoles in both acquisition modes. D-FF-OTT images of *Candida albicans* from Figure 2c,d show that typical yeast structures such as plasma and nuclear membranes (white arrow and empty arrowhead) and organelles (white-filled arrowhead) have distinct dynamic signals. Briefly, D-FF-OTT shows a bright orange signal for yeast membranes (white arrows), while nuclear membranes have a less intense yellow/green signal (empty white arrowheads), and an intense pink signal is attributed to organelle dynamics inside fungi cytosol (white arrowheads). Figure 2e and f show bright-field and FF-OTT images of *Candida parapsilosis*, respectively, while Figure 2g,h show D-FF-OTT images. We observe typical budding phenomena and pseudohyphae, and we can easily distinguish structures such as vacuoles (white asterisk), plasma and nuclear membranes (white arrow and empty triangles), and organelles (white-filled arrowhead).

As for Figure 2d, Figure 2h shows a similar bright orange signal for yeast membranes (white arrows), a less intense yellow/green signal for nuclear membranes (empty white arrowheads), and an intense pink signal from organelles inside the cytosol (white arrowheads). In *Candida parapsilosis*, we also observed that while vacuoles can hardly be seen in bright-field images, they appear more clearly in FF-OTT. Vacuoles show smaller light scattering than the cytosol in FF-OTT (Figure 2f) and appear as dark compartmentalized regions in which no dynamic signal is observed, as shown in Figure 2h (white asterisk).

Figure 3 shows images of *Aspergillus fumigatus*. Note that D-FF-OTT imaging was recorded without prior staining. Figure 3a–d shows a conidium constituting of a stipe and a vesicle with phialides. The unstained and blue-stained conventional bright-field images (Figure 3a and b, respectively) display similar structures as those observed within FF-OTT and D-FF-OTT images (Figure 3c and d, respectively). If we can only guess the presence of organelles in stipes (Figure 3a–c), Figure 3d reveals a clear pale blue dynamic signal that is specific to these intracellular compartments. We also observed branched hyphae that are not homogeneously metabolically active; we see both areas that appear full of vesicles and organelles (Figure 3h) and empty metabolically inactive sections (as seen in the lower part of Figure 3c,d,g,h). We also distinguish septa in bright-field microscopy (white arrowheads, Figure 3e) and FF- and D-FF-OTT (white arrowheads, Figure 3g,h). The dynamic signal of membranes is low (faint green to orange), as well as the one from septa, compared to organelles (intense pink to white). Similar characteristics were observed in *Aspergillus niger* (Figure 4).

Figure 4 shows different parts of *Aspergillus niger*, namely stipes, vesicles, phialides (Figure 4a–d), and hyphae (Figure 4e–h). FF-OTT and D-FF-OTT (Figure 4c,d) allow for the same observation as bright-field microscopy of unstained and blue-stained conidia regarding the structure of the conidial head (Figure 4a and b, respectively). We did not observe any dynamic signal in vesicles, while phialides and conidia display either a weak green dynamic signal, close to background noise, or a brighter pink signal (Figure 4d). As the conidial head is large and mostly out of the imaging plane, we imaged at different depths, showing the relative consistency of this signal (Appendix A). Hyphae can be seen in bright-field microscopy (Figure 4e,f), as well as in FF- and D-FF-OTT (Figure 4g,h). Similar to *Aspergillus fumigatus*, some hyphae are vacuolated while some are not, and one can distinguish intracellular organelles in all images and some septa (white arrowheads, Figure 4e–h) that appear with an intense red color, meaning they are highly dynamic. Moreover, D-FF-OTT allows for the observation of an intense dynamic signal in the cytosol of hyphae.

Figure 5 shows typical *Rizhopus arrhizus* structures. Figure 5a–d shows a rhizoid which consists of smooth-walled hyphae with some sporangiospores (white arrowheads). *Rizhopus arrhizus* hyphae are sparsely or non-septate, as are other Mucorales [20]. D-FF-OTT shows very similar images to what is seen in conventional bright-field microscopy, adding the dynamic intracellular signal as seen in Figure 5d; this new modality shows a rather weak signal in both membranes and septa, with a global color ranging from faint green to pale orange, as opposed to multiple hyphae organelles displaying intense and saturated dynamic signals from pink to white depending on the intracellular compartment. Figure 5e–h shows sporangiophores, columellae, and sporangiospores. Again, bright-field microscopy allows for the observation of intracellular organelles (Figure 5e) but D-FF-OTT adds a specific signal to these structures (Figure 5h). In Figure 5g, we clearly distinguish that sporangiospores are angular and ellipsoidal, with striations of the surface (white arrowhead).

All D-FF-OTT images that are presented were realized with fungi that had not been stained with lactophenol blue. Indeed, after blue staining, no dynamic signal was seen in fungi, which is consistent with the toxicity of lactophenol blue that has fixation properties (Appendix A).

### 3.2. Quantitative Analysis of D-FF-OTT Images of Candida albicans

Previous results have shown a raw description of the images acquired in FF-OTT and D-FF-OTT, enabling the differentiation between fungal species. However, FF-OTT and D-FF-OTT carry rich and possibly specific information about the intracellular dynamics of these cells. We now want to investigate whether D-FF-OTT can be used for the automated classification of fungal compartments and further on, of different fungal species using automated segmentation tools and a quantitative analysis of the intracellular fluctuations.

Figure 6a–f indicates for each channel of the dynamic image (hue, saturation, and brightness) the fungal structures we wanted to quantify signal properties for. The white arrows show yeast membranes, white empty triangles indicate nuclear membranes, and white-filled arrowheads point out organelles for each of the three images that are used to realize the dynamic image (Figure 6a,c,e). The figure also displays the resulting regions of interest (ROIs) and masks that are segmented by the algorithm; the yeast membrane ROI is visualized in blue, while intracellular compartments are highlighted in yellow. Among these intracellular compartments, the nuclear membrane ROI is identified by an orange circle, while the organelle is delimited by a purple line. We have developed an automated segmentation algorithm based on YOLOv8 to extract pixels corresponding to different fungal compartments. Altogether, we managed to segment the cytoplasmic membrane (*n* = 320), the nucleus membrane (*n* = 188), and intracellular organelles (*n* = 135) of *Candida albicans* (Figure 6b,d,f). Each histogram shows the normalized value for a different parameter of the dynamic image (central fluctuation frequency, frequency range, and amplitude of fluctuations) within these three ROIs (plasma membranes, nuclear membranes, and organelles, in blue, orange, and purple, respectively) for the corresponding property.

In Figure 6a, the plasma membrane appears brighter than the nuclear membrane (white empty triangle), and organelles are darker (white arrowhead). Accordingly, greyscale values that are plotted in the corresponding graph (Figure 6b) are higher for the plasma membrane and lower for organelles. Student’s *t*-test showed that these three structures have significantly different central fluctuation frequencies (*p* < 0.0001). Figure 6c,d shows a similar analysis for the frequency range and Figure 6e,f presents the results for the amplitude of fluctuations. All structures had significantly different signals (*p* < 0.0001) for each spectral property except plasma and nuclear membranes for the amplitude of fluctuation (*p* = 0.14), as specified in Figure 6f.

## 4. Discussion

Here, we illustrate the potential application of D-FF-OTT to medical mycology, revealing its efficiency to properly image specific structures of several types of fungi. For all considered fungal strains, FF-OTT images showed similar or even better performances than bright-field standard microscopy. Moreover, D-FF-OTT brought an additional feature that could be correlated to intracellular fungal metabolism. This modality gives us information about the dynamics of tissues we are observing, and this vibration is linked to the physical properties of living tissues and the intracellular transport within them.

For instance, the vibration characteristic of membranes will depend on their stiffness and their physical environment. Thus, we expect that each type of membrane may have a different D-FF-OTT signature, as their fluctuation properties depend on these physico-chemical properties [21,22,23]. Fungal cell walls are specific, as their plasma membranes are surrounded by a polysaccharidic layer that is necessary to viability and virulence. This may represent an opportunity in fungal infection diagnosis, as fungal membranes may have a dynamic signal that is different from human cell membranes, and AI-based recognition algorithms would be of great help for such a diagnostic tool.

The diversity in the intracellular membranes of yeasts has been documented, leading to different physical properties such as an increased rigidity of the plasma membrane compared to mitochondrial or nuclear membranes [24]. If we consider Figure 2, we can see that the plasma membrane signal is red/orange, while nuclear membranes appear blue/green, indicating that D-FF-OTT mean frequencies are higher for plasma membranes than for nuclear membranes. This is in accordance with a higher rigidity of plasma membranes. However, D-FF-OTT does not yet possess the spatial resolution to observe the signal of mitochondria or lipid body membranes. Cell walls and septa of *Aspergilli* and *Rhizopus* also appear red/orange (Figure 3, Figure 4 and Figure 5), while intracellular bodies have white, blue, or pink signals, suggesting a higher rigidity of plasma membranes and fungal walls compared to intracellular structures such as vacuole membranes. Vacuole content still remains void of signal in both observed *Aspergillus* species, as well as in *Candidae*, which is consistent with the fact that although vacuoles are complex organelles, they are homogeneously constituted of fluids [25]. Lastly, the composition of the cell walls of conidia and hyphae of *Aspergilli* is different [26], which may explain the difference in D-FF-OTT signals between those compartments, as shown in Figure 4d,h. Globally, membranes have a signal that appears less bright than organelles, meaning that the amplitude of variation in their vibrations is lower, which is confirmed by the analysis performed in Figure 6f. Fungal cell walls have a key role in many physiological functions such as cell structure, division, and metabolism, as well as pathogenic processes such as adhesion and invasion [27]. Three out of four classes of antifungal medication target fungal cell walls or membranes, namely echinocandins, polyenes, and azoles. Thus, the treatment of fungi with these substances may alter their walls or plasma membranes and modify their physico-chemical properties, which may lead to a change in the D-FF-OTT signal. This has not been investigated yet, but it may represent a way of using D-FF-OTT as a new screening method for testing antifungal molecules.

As intracellular organelles have dimensions inferior or close to a few microns, D-FF-OTT is a powerful tool to observe these structures without staining, bringing new insights into the intracellular activity of fungi [28,29,30,31]. In our situation, we observed structures with a dimension above a quarter of the LED wavelength. In this case, light beams are mostly scattered forward (Mie scattering), thus enabling their interference with the fraction of the beam that is not scattered by the object. We can consider different compartments, those which appear static, such as the nucleus and vacuoles, and those which are obviously moving on a short time scale (<1 s). For static compartments, we have demonstrated how D-FF-OTT could return not only quantitative information regarding membrane dynamics, as for nuclei in Figure 6, but could also unveil fluctuation dynamics from the inside of these compartments. Given the number, shape, and localization of fast-moving organelles, it is likely that the pink intense signal from Figure 2d,h corresponds to mitochondria and lipid bodies [28,30]. D-FF-OTT allows for the characterization of their intracellular distribution and trafficking and reveals distinctive fluctuation properties that to our knowledge have never been described before. These organelles can be considered as colloidal particles (with sizes that are close to 1 μm) and their movement may be confined to Brownian motion [32]. In this case, parameters influencing their dynamics would merely be their sizes and the viscosity of cytosol. Thus, our experiments may constitute a way of gaining insight into the intracellular properties of these cells. Again, it is also important to underline that these images can be acquired in a very short time without prior staining and require living organisms, while fungal characterizations often require long and complex microscopy or prior staining processes [31,33] and is performed on fixed samples.

Finally, the clinical need of a fast and unambiguous identification of fungi responsible for an IFI is of primary importance and artificial intelligence (AI)-based programs may help achieve this goal. AI has already been successfully applied to cytology to help speed up and improve diagnosis [12], and efforts to translate these techniques toward fungal cytology are being undertaken [5,34,35]. We first developed an algorithm that was able to differentiate *Candida albicans* and *Candida parapsilosis* based on their shape differences (Appendix A) and then implemented a second algorithm able to segment intracellular compartments of *Candida albicans* (Figure 6). This allowed us to show that these compartments had significantly different signals in D-FF-OTT, revealing new perspectives in fungal metabolism knowledge and AI-assisted diagnosis. By associating cell shape metrics with intracellular compartment number, size, type, and local dynamics, this gives several possible metrics to achieve a specific automatic identification of fungi type and virulence. This obviously requires more data and fungal types, which was beyond the scope of this study but will be the object of future work. Such a technology would be of great importance in medical mycology, as it may importantly decrease the diagnosis delay of IFIs, with potential significant outcomes for patients.

## 5. Conclusions

This preliminary work not only demonstrates that D-FF-OTT allows for direct fungal imaging without prior staining but also opens perspectives to better understand the metabolic activity or intracellular trafficking of various types of fungi. Concomitant to advances in experimental mycology or mechanistic studies, such a technique may greatly improve the clinical management of IFI. Indeed, clinicians and microbiologists are in need of novel diagnostic tools that will enable them to speed up IFI diagnostics with improved sensitivity. Moreover, the associated development of innovative AI-based recognition algorithms will participate in the enhancement of the specificity of this tool. As we demonstrated that a simple algorithm was able to identify specific areas of a yeast and quantify their fluctuation properties, revealing new perspectives in automatized fungi recognition from D-FF-OTT images, we are confident that the optimization of this technique will allow us to characterize more precisely the specific signal of each type of intracellular organelle, and that the use of AI-based segmentation could help develop new diagnostic tools with improved specificity and rapidity, with a potential impact on IFI diagnostics and care.

## Figures and Tables

**Figure 1 jof-10-00741-f001:**
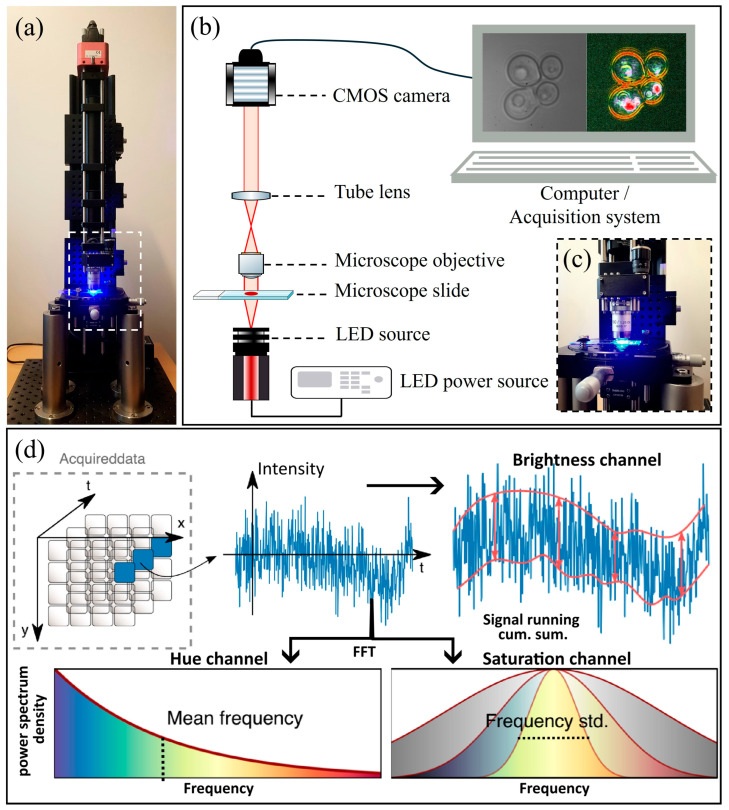
Illustration (**a**) and schematic representation of the prototype OTT system (**b**), focus on the objective area of the microscope (**c**), and a schematic explanation of signal treatment for dynamic imaging reconstruction (**d**) during *Candida albicans* imaging. The computer window in (**b**) shows both structural greyscale (left) and dynamic (right) modes. (Adapted from Ref [17]).

**Figure 2 jof-10-00741-f002:**
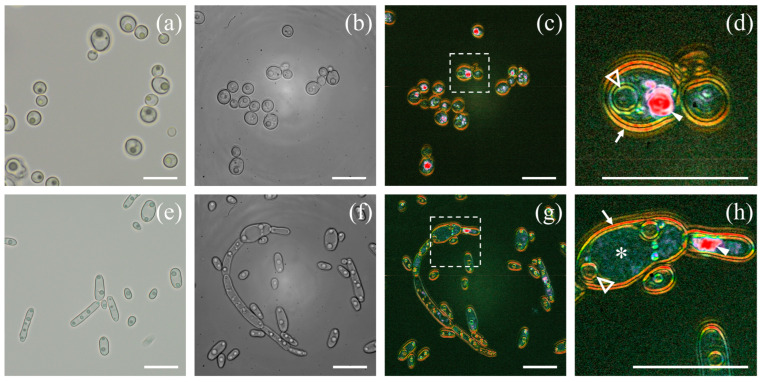
Bright-field and D-FF-OTT imaging of *Candida albicans* (**a**–**d**) and *Candida parapsilosis* (**e**–**h**). Bright-field imaging of *Candida albicans* (**a**) and *Candida parapsilosis* (**e**). FF-OTT (**b**,**f**) and D-FF-OTT (**c**,**d**,**g**,**h**) of *Candida albicans* (upper row) and parapsilosis (lower row). Dashed squares in Figure 2 (**c**,**g**) are shown with a higher magnification in (**d**,**h**). White arrows indicate cell membranes, empty triangles indicate nuclear membranes, and white-filled arrowheads point out organelles. Scale bars represent 10 μm.

**Figure 3 jof-10-00741-f003:**
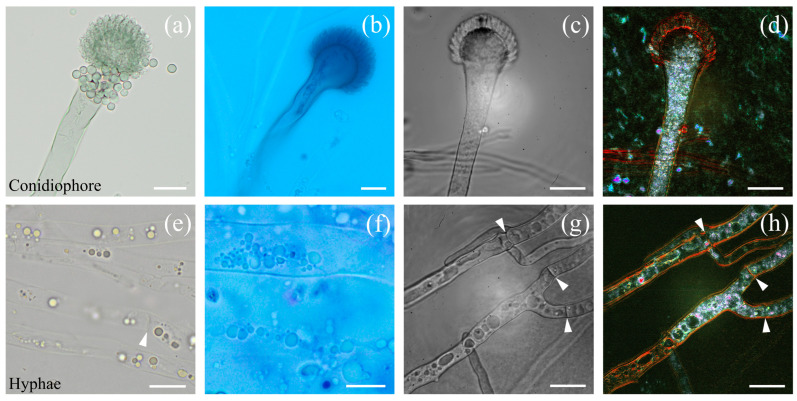
Bright-field and D-FF-OTT imaging of *Aspergillus fumigatus*. Bright-field imaging of unstained (**a**,**e**) and lactophenol blue-stained (**b**,**f**) fungi. FF-OTT (**c**,**g**) and D-FF-OTT (**d**,**h**) of similar structures. White arrowheads point out septa. Scale bars represent 10 μm.

**Figure 4 jof-10-00741-f004:**
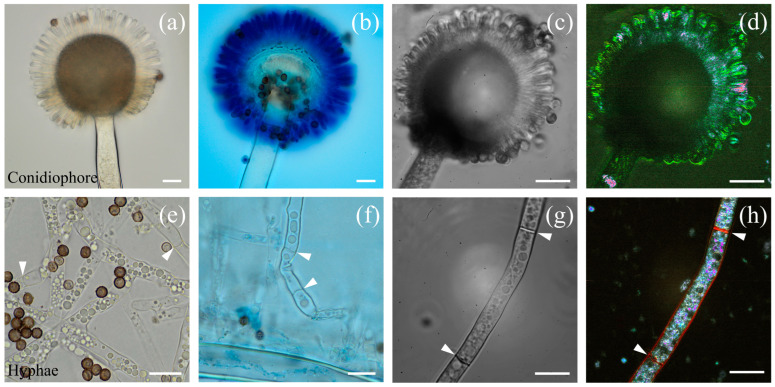
Bright-field and FF-OTT imaging of *Aspergillus niger*. Bright-field imaging of unstained (**a**,**e**) and lactophenol blue-stained (**b**,**f**) fungi. FF-OTT (**c**,**g**) and D-FF-OTT (**d**,**h**) of similar structures. White arrowheads point out septa. Scale bars represent 10 μm.

**Figure 5 jof-10-00741-f005:**
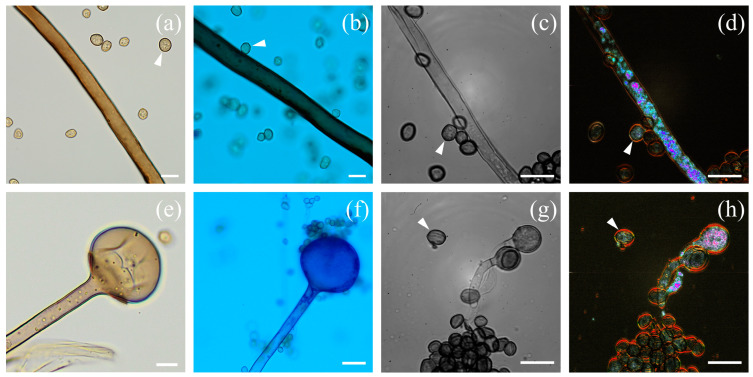
Optical and FF-OTT imaging of *Rhizopus arrhizus*. Optical imaging of unstained (**a**,**e**) and lactophenol blue-stained (**b**,**f**) fungi. FF-OTT (**c**,**g**) and D-FF-OTT (**d**,**h**) of similar structures. White arrowheads indicate sporangiospores. Scale bars represent 10 μm.

**Figure 6 jof-10-00741-f006:**
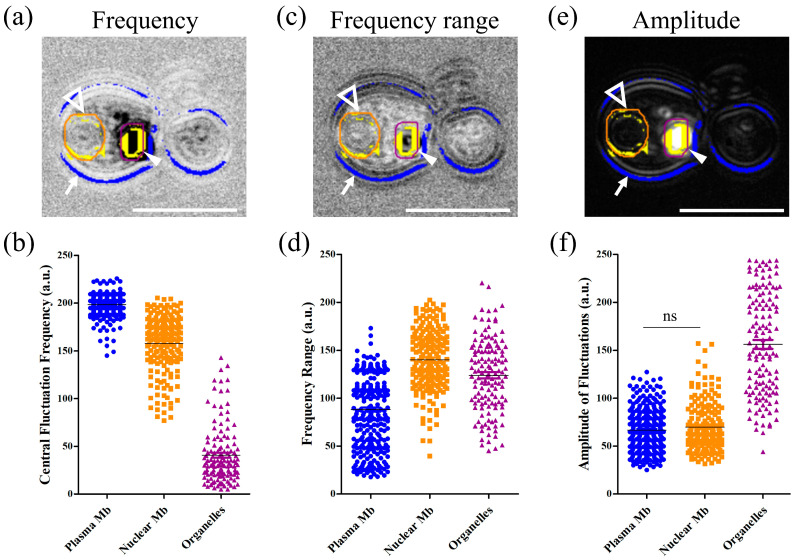
Quantitative spectral analysis of cellular structure of *Candida albicans*. D-FF-OTT registers three spectral parameters that are the central fluctuation frequency (**a**,**b**), frequency range (**c**,**d**), and amplitude of fluctuations (**e**,**f**). Images were analyzed and signals were quantified for each spectral characteristic for three distinct structures, namely the plasma membranes (blue circles), nuclear membranes (orange squares), and lipid droplets (purple triangles). Statistical analysis showed a significant difference (*p* < 0.0001) for any comparison except for the comparison between the amplitude of fluctuations of plasma and nuclear membranes, as indicated on corresponding graph (ns in (**f**)). White arrows indicate cell membranes, empty triangles indicate nuclear membranes, and white-filled arrowheads point out organelles. Scale bars represent 10 μm.

## Data Availability

The original contributions presented in the study are included in the article/Appendix A; further inquiries can be directed to the corresponding author.

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
