# Peer review of "Label-Free Optical Transmission Tomography for Direct Mycological Examination and Monitoring of Intracellular Dynamics"

_jof, 2024, doi:10.3390/jof10110741_

Round 1

Reviewer 1 Report

The authors report about a study on label-free optical transmission tomography for direct mycological examination and monitoring of intracellular dynamics. After an explanation of the underlying principles, experimental results from Candida albicans, Aspergillus fumigatus, or Rhizopus arrhizus are presented.

In general, the manuscript is motivated, organized and includes adequate references. The experimental investigations appear to be accurately performed. The presented results are  original and plausible. The authors address an important topic in mycology: Label-free quantitative analysis , which may be of interest for the area of fungal infections diagnostics and label-free quantitative biomedical imaging. In summary, the content of the manuscript appear to be suitable for the Journal of Fungi.

1.      Method (main point):

a. The applied imaging method is denoted as dynamic full-field optical transmission tomography (D-FF-OTT). However, the tomography feature of the method seems to be not utilized in the presented study. The authors should clarify the text in the title, abstract, the introduction and in the methods description concerning this topic.

b.  From the manuscript descriptions, it becomes not clear how the presented experimental setup can be utilized for tomographic imaging. The authors should clarify this topic.

c. From Fig. 1b it seems that the experimental setup is highly similar (identical?) to a common bright field microscope. The authors should describe and emphasize possible differences/novelty aspects of their approach in comparison with a bright field microscopy setup with more details.

2.      Methods/results:

a.  The authors should add information about the required computer hardware for image processing and the time duration for training of the applied convolutional neural networks.

3.      Discussion:

a.  With view to the intended application of the proposed method in diagnostics the authors should briefly discuss the possible impact of system alignment and sample preparation procedures on the measurement data/data evaluation and possibly resulting demands on standardization regarding regulatory issues.

Reviewer 2 Report

The authors present their work on "Label-free optical transmission tomography for direct mycological examination and monitoring of intracellular dynamics." The research is based on an advanced analysis of temporal datasets acquired using customized bright-field transmission microscopy. The experimental setup is straightforward. The analysis relies on FFT analysis of the temporal datasets, with colors in HSV (Hue, Saturation, Value) assigned to the images. The authors demonstrated their method's performance across various samples (Fig. 2-5).

1. My primary concern is understanding the physical significance of the colors used. Why was HSV chosen over RGB? What do the red, green, and blue colors represent the physical process in this context?

2. There appears to be a gap between Fig. 1 and Fig. 2. It is recommended to include a figure that describes the data post-processing flowchart and the coloring mechanism.

3. The results presented in Fig. 2-5 primarily illustrate the differences between bright-field microscopy and the authors' method. It is suggested to design experiments with underlying biological changes, such as the effects of overheating or drug treatments that cause fungi to die, activate, or deactivate, and to show the quantitative changes in the colors.

1. Line 110: It appears that Fig. 1 is incomplete, as it references "The computer window shows both structural greyscale (left) and dynamic (right) modes." However, the left and right parts of the computer window are not clearly depicted. Could you clarify where these parts are located in the figure?

2. Line 203:

(1) There is a lack of quantitative comparison between the results in Fig. 2-5.

(2) Additionally, a zoom-in on (b) and (f) would be helpful.

(3) Why are there fringes surrounding the fungi in (d, h)?

3. Line 217: Upon close inspection of (g) and (h), one can observe that certain details, such as the edges of organelles and septa, are more clearly defined in (g) than in (h). The same observation applies to the results shown in (g)-(h) on Line 235 and Line 266. Could the authors explain why?

4. This suggests that the results in Fig. 2-5 do not strongly demonstrate the advantages of D-FF-OTT over FF-OTT in visualizing intracellular structures. I would argue that they both have their own advantages and limitations. 

Round 2

Reviewer 2 Report

I think the authors adequately addressed my previous concerns, I don't have further comments.

I think the authors adequately addressed my previous concerns, I don't have further comments.